# Bioengineering of Extracellular Vesicles: Exosome-Based Next-Generation Therapeutic Strategy in Cancer

**DOI:** 10.3390/bioengineering8100139

**Published:** 2021-10-10

**Authors:** Priyanka Saha, Suchisnigdha Datta, Sukanya Ghosh, Anurima Samanta, Paramita Ghosh, Dona Sinha

**Affiliations:** Department of Receptor Biology and Tumor Metastasis, Chittaranjan National Cancer Institute, 37, S.P. Mukherjee Road, Kolkata 700026, India; poojasaha.saha79@gmail.com (P.S.); ronie.bhu@gmail.com (S.D.); sukanyaghosh19091981@gmail.com (S.G.); anurimasamanta@gmail.com (A.S.); paromitaghosh02051995@gmail.com (P.G.)

**Keywords:** exosome, bioengineering, immunotherapy, exosomal cargo delivery, recombinant proteins, ncRNA, chemotherapy

## Abstract

Extracellular nano vesicles and exosomes hold compelling evidence in intercellular communication. Exosomal intracellular signal transduction is mediated by the transfer of cargo proteins, lipids, micro (mi)RNAs, long noncoding (lnc)RNAs, small interfering (si)RNAs, DNA, and other functional molecules that play a pivotal role in regulating tumor growth and metastasis. However, emerging research trends indicate that exosomes may be used as a promising tool in anticancer treatment. This review features a majority of the bioengineering applications of fabricated exosomal cargoes. It also encompasses how the manipulation and delivery of specific cargoes—noncoding RNAs (ncRNAs), recombinant proteins, immune-modulators, chemotherapeutic drugs, and other small molecules—may serve as a precise therapeutic approach in cancer management.

## 1. Introduction

Specialized nanoparticles, exosomes, have gained considerable attention from researchers and clinicians by virtue of their intercellular communication and efficient drug delivery property [1]. Exosomes are highly advantageous for therapeutic purposes due to their high stability, targetability, less immunogenicity, prolonged half-life, and ability to cross the blood–brain barrier (BBB) [2]. Exosomes may be modified with several molecules, chemotherapeutic drugs, functional proteins, and genetic materials, which might emerge as a potential next-generation anti-cancer strategy. Exosomes may invade physiological barriers, which were generally impenetrable by other synthetic drug delivery vehicles. This potential ability of exosomes have fascinated us to review several therapeutic strategies that may improve cancer treatment. They may be utilized for next-generation diagnostics, in monitoring several disease progressions and their accurate therapy [3]. They can even act as an excellent alternative for stem cell therapy [4]. However, the clinical applications of exosomes are limited to date, making it an area of greater interest. The challenge lies in their efficient separation, characterization, and detection with specific biomarkers. Once the barriers in the arena of exosomes are tackled, they may act as the most efficient vehicle for carrying molecules to facilitate cancer therapeutics [5]. Therefore, future research is warranted to overcome these challenges. Nowadays, exosome-based liquid biopsy helps to determine the prognosis of cancer patients and other diseases [6]. The present review has envisaged several therapeutic strategies that might be useful for future pre-clinical and clinical research. 

Dai et al. have reviewed the role of exosomes in cancer, mainly focusing on the several components of exosomes and how they may be related to tumor progression [7]. Others have either emphasized source-specific exosomes such as tumor-derived exosomes (TEXs) [8] and human breast milk exosomes [9] or the utility of exosomes in therapeutic strategies against a specific cancer such as breast cancer [10]. On the contrary, the present review has tried to provide insight into the role of exosomes in the regulation of cancer, the strategies of exosomal bioengineering, and their implementation for future anticancer treatment against all cancer types. The wide array of exosome delivery modalities, the therapeutic implications of exosomes involving ncRNAs, immune modulations, chemotherapeutic drugs, natural phytochemicals, small molecules, recombinant proteins, and the emerging concepts of fusogenic exosomes and vexosomes have been comprehensively reviewed, which might be interesting realms of future research and therapeutic strategies.

## 2. Biogenesis, Structure, and Composition of Exosomes

Exosomes are nano-sized, anucleated, spherical bilayer structures with a cup-shaped morphology and average diameter of 30–100 nm [11]. Exosomes are formed by inward budding of the cell membrane containing ubiquitinated surface receptors leading to the formation of early endosomes [12]. These early endosomes become late endosomes and intraluminal vesicles with the help of the Golgi apparatus. Intraluminal vesicles accumulate in the endosome, leading to the formation of multivesicular bodies. The fusion of multivesicular bodies with the plasma membrane results in the release of internal vesicles into the extracellular space by several RabGTPase [13]. Extensive studies have revealed a detailed mechanism of exosomal biogenesis. During the maturation of endosomes or multivesicular bodies, intraluminal vesicles are formed within the lumen of the organelles [14]. It involves two main types of machinery: an endosomal sorting complex required for transport (ESCRT)-dependent pathway and an ESCRT-independent pathway. ESCRT involves four protein complexes (ESCRT-0, -I, -II, and -III) and associated proteins VPS4 ATPase that are responsible for the recycling of exosomes. ESCRT-I and -II enable bud formation, while ESCRT-III along with Alix is responsible for vesicle scission. ESCRT-0 mainly drives cargo clustering in an ubiquitin-dependent manner. ESCRT-independent machinery involves chaperons; tetraspanin; and lipids such as cholesterol, proteolipid proteins, phospholipase, D2, etc. [15]. Certain tumor cells have been reported to secrete exosomes with phosphatidylserine on their membrane [16]. In cancers, several components and pathways responsible for exosomal biogenesis and secretion are intensely regulated [17]. For example, Rab family proteins such as Rab27a, Rab27b, Rab5, Rab11, and Rab35 are responsible for early sorting, maturation, and recycling of endosomes and often remain constitutively active in cancer cells [18,19,20]. In rat adenocarcinoma cells, the mRNA and protein composition of secreted exosomes were modified by one of the tetraspanins, TSPAN8 [21]. Tetraspanin CD63 was found to be responsible for sorting of a melanosomal protein and membrane invagination in exosomes secreted from human melanoma cells in an ESCRT-independent manner [22]. A better understanding of exosome biogenesis and secretion machinery may help to develop new therapeutic strategies.

Exosomes contain mRNA, lipids, and ncRNAs as well as both cytosolic and membrane proteins [12]. Unlike the cytoplasmic membrane, the exosomal membrane shows a balanced composition of phospholipids consisting of phosphatidylcholine, phosphatidylethanolamines, phosphatidylinositol, phosphatidylserine, and sphingomyelin in the ratios 43:23:12:12:9 and 26:26:9:19:20, respectively. The cholesterol concentration in exosomes is similar to that in the cytoplasmic membrane, but diacylglycerol is reduced to 50% [23]. Proteins in exosomes include heat shock proteins (Hsp70 and Hsp90) and those that are required for fusion with the target cells [GTPases, annexins, flotillin, and cell targeting protein tetraspanins (CD9, CD63, CD81, and CD82)] [24]. Exosomes are also rich in miRNA and mRNA. Exosomes that are secreted by astrocytes and glioblastoma cells are even rich in mitochondrial DNA [12]. Based on this structural arrangement and properties, exosomes are developed for therapeutic implications against various diseases, especially cancers.

## 3. Exosomes in Cancer Regulation

### 3.1. The Protumorigenic Activity of Exosomes

Exosomes are involved in every aspect of tumor progression such as immune evasion, a gain in migratory and invasive capacity, angiogenesis, cancer tissue enlargement, and ultimately metastasis [25]. They can act as a vector for the carriage of several molecules and genetic materials from donor to recipient cells.

Secreted microvesicles from hypoxic glioblastoma cells released tissue factors that activated hypoxic endothelial for hyperplasia and hypercoagulation [26].

Exosomes derived from different cancer cells were also associated with the activation or inhibition of immune cells. As reviewed by Osaki et al., colon cancer-derived exosomes expressed Fas ligand, which caused T cell apoptosis, and breast cancer cell-derived exosomes blocked natural killer (NK) cell activation by blocking interleukin (IL)-2 [25]. Pancreatic cancer cell-derived exosomes inhibited immune response via miR-203 and thus downregulated Toll-like receptors, and downstream cytokines such as tumor necrosis factor-alpha (TNF-α) and IL-12 in dendritic cells (DC) [27].

The fibroblast-secreted exosome component CD81 along with Wnt-planar cell polarity signaling in breast cancer cells induced protrusive activity and enhanced metastasis and motility [28]. Pancreatic ductal adenocarcinoma-derived exosomes were observed with a high expression of the macrophage migration inhibitory factor, which promoted a premetastatic niche in liver and metastasis at a later stage [29]. Other exosomal molecules such as Apolipoprotein E [30], HSP70 [31], Wnt4 [32], epidermal growth factor receptor (EGFR) [33], and integrin αVβ6 [30] were reported to be involved in tumor progression in the recipient cells. Several exosomal ncRNAs are emerging as prominent players in tumor progression. MiRNAs such as colorectal cancer cell-derived exosomal miR-934 interacted with tumor-associated macrophages and induced premetastatic niche formation via the polarization of M2 macrophages and ultimately caused colorectal cancer liver metastasis [34]. In another study, exosomes derived from highly metastatic human oral cancer cells were found to transfer two onco-miRs, miR-1246 and miR-342-3p, to poorly metastatic cells at adjacent or distance sites and induced increased cell motility and invasive capacity [35]. Exosomal miRNAs such as miR-663b [36], miR-21 [37], miR-105 [38], miR181C [39], miR-106 [40], and miR-222 [41] and other lnc RNAs such as Sox2ot [42], ZFAS1 [43], and HOTTIP [44] promoted tumor migratory properties in several cancer types.

Donor hepatocellular carcinoma (HCC)-derived exosomes transferred Lysyl-oxidase-like 4 between HCC cells to human umbilical vein endothelial cells (HUVECS), where they promoted angiogenesis and cell migration in a paracrine manner [45].

### 3.2. The Antitumorigenic Activity of Exosomes

Despite having several pro-tumor effects, exosomal cargoes are also involved in inhibiting tumor progression. 

Exosomal constituents exhibited antitumor responses via immune modulation [46]. A study on NK cell-derived exosomes previously exposed to neuroblastoma cells exhibited antitumor properties [47].

Normal cell-derived exosomes transferred long ncRNA (lncRNA) PTENP1 to bladder cancer cells, which reduced tumor progression both *in vitro* and *in vivo* [48]. Other exosomal miRNAs such as miR-144 [49] and miR-520b [50] inhibited non-small cell lung cancer (NSCLC) progression through the downregulation of cyclin E1 and E2 migration of pancreatic cancer cells, respectively. Exosomal miR-497 suppressed the migratory properties of lung cancer cells via the inhibition of growth factors and cyclin E1 [51]. Even circulating RNA circ-0051443 carried by exosomes suppressed tumor progression in HCC cells [52]. Exosomal miR-375 inhibited cell proliferation and the invasive properties of colon cancer cells [53]. 

Apart from miRNA and lncRNA, other exosomal molecules such as gastrokine 1 inhibited gastric carcinogenesis [54]. Exosomal miR-139 derived from cancer-associated fibroblasts inhibited gastric cancer progression by suppressing matrix metallopeptidase 11 expression [55]. Therefore, exosomal cargoes that are involved in tumor suppression may be beneficial for the anticancer therapeutic approach. 

## 4. Exosomes—A Tool in Cancer Management

Exosomal constituents give a miniature reflection of their parental cells. Cancer cells produce a significantly greater number of exosomes or TEXs, making them ideal for precise detection. The exosome is a compact nanovesicle stably containing the antigenic and genomic information, ensuring its role as a reliable and early cancer biomarker. Being non-living and easy to handle, exosomes are emerging as a promising intercellular communication tool to find a sustainable cure for cancer [17]. According to the level of organizational complexity and biological applications, the exosomes are interestingly more bio-functional and heterogeneous than simple antibodies, RNAs, or synthetic compounds and at the same time easily manipulatable in comparison with cells, tissues, or organs because of their low engineering difficulty [56]. The above-mentioned unique properties of exosomes ascertain the bright prospect of exosomal bioengineering in cancer diagnostics and therapeutics. Different techniques for exosome isolation and incorporation have been summarized in Table 1 and Table 2.

### 4.1. Exosomal Isolation Methods

Depending on the cell source, exosomes are used for experimental, diagnostic, or therapeutic engineering purposes and can be obtained from divergent sources. For example, mesenchymal-stromal cell (MSC)-derived exosomes may arise from various origins such as pulmonary, renal, hepatic, neurological, muscular, adipose, hematopoietic, and cardiac tissues [57] and can be found in any bodily fluids or ascites, and extracellular media. Other than these, plant-derived exosomes (fruit or vegetable) and milk-derived exosomes have also shown anti-tumor potential. Some of the prevalent exosome isolation methods have been described below.

#### 4.1.1. Ultracentrifugation

Ultracentrifugation is the most practiced method and is known as ‘the gold standard’ for isolating exosomes. It requires a series of gradually increasing centrifugal speeds, which allows for separation of different organelle compartments at a different speed and finally ultracentrifugation to obtain the exosomal fraction. However, this repetition of centrifugation may be the reason for the lower yield, damage to the vesicular integrity, and other macromolecular contaminations. Therefore, clubbing other methods such as density gradient separation using sucrose or iodixanol along with ultracentrifugation may be a solution [58].

#### 4.1.2. Ultrafiltration

By using a set of membranes of different porosity, exosomes are concentrated by filtering out other macromolecules. Though the yield in this process is better than that of ultracentrifugation, the mechanical pressure may rupture the vesicle. Moreover, the membrane adherence property of the exosomes may lead to poor recovery after separation. Tangential flow filtration, which is suitable for mass isolation, uses superfine pore size for higher yield and lower contamination of exosomes [59].

#### 4.1.3. Size Exclusion Chromatography

Size exclusion chromatography involves a pure fraction of small-sized macromolecules such as exosomes, which may be eluted from the rest as they have higher retention times inside the porous beads of column depending on the gravity flow. The gravity flow usually varies with the pore size. Due to its ability to maintain exosomal integrity and functionality, size-exclusion chromatography is an ideal method of isolation from small volume (e.g., diagnostic purpose) but is not suitable for large-scale purification (e.g., therapeutic purpose) [60].

#### 4.1.4. Immuno-Affinity Capture

Immuno-affinity capture is a method that utilizes immuno-affinity for exosomal surface markers (e.g., EpCAM, CD9, CD63, and CD81) on its membrane, and thus, a particular subset of exosome can be extracted with the help of their corresponding antibodies. However, the inherent heterogeneity of bodily fluids is a practical limitation towards the success of this method. Techniques such as microfluidic chips or magnetic beads coated with a specific antibody are excellent but expensive methods of exosome isolation that are better than ultracentrifugation [61].

#### 4.1.5. Polymer-Based Precipitation

Polymer-based precipitation is the natural tendency of the free exosomes to become wrapped around any favorable inert surface. It can be enhanced by using adsorbing surfaces made of commercially available polymers (e.g., quick or polyethylene glycol). Though this method has the drawback of contaminating non-exosomal precipitants, it is better for maintaining the integrity and higher yield of exosomes. This simple yet fast precipitation method is best suited for isolation of the whole exosome or its components such as RNA or proteins [62].

**Table 1 bioengineering-08-00139-t001:** Methods of exosome isolation: different methods of exosome isolation along with their advantages and disadvantages.

Method	Principle	Advantage	Disadvantage	Reference
Ultracentrifugation	The constituents are separated based on their density and size	High yielding capacity, cost effective, low risk of contamination	Damage prone due to high speed, requires special equipment, time consuming	[58]
Ultrafiltration	Different exosomes are separated based on their size	Fast, cost effective, no need for special equipment, reduced labor	Low purity	[59]
Size exclusion chromatography	Different exosomes are separated based on their size	High purity, biological activity is preserved	Moderate cost, requires special equipment	[60]
Immunoaffinity capture	Exosomes are separated based on their membrane-bound protein and receptors.	High purity, isolation of ligand specific exosome	Specific ligands need to be established, yield and capacity are low, receptor may be blocked	[61]
Polymer-based precipitation	Exosomes are precipitated using a water excluding polymer	Possibility for kit-based isolation, user friendly, no requirement of special equipment	Risk of contamination is high, similar to proteins.	[62]

### 4.2. Exosomal Incorporation Methods

The natural origin of exosomes render their safety from a bodily immune attack such as phagocytosis and macrophage-mediated elimination [63]. Therefore, they may be available in circulation for an extended period and can become internalized whenever and wherever possible. Exosomes have a benefit over other microvesicles because, on internalization, they do not become degraded by the intracellular lysosomal trafficking system and stay stable inside the cytoplasm [63]. Moreover, this uptake is very much dose- and time-dependent [64]. Heparin treatment or dynamin blocking can substantially inhibit the process [65].

Though the donor and the recipient tissues/cells are highly specific, the regulatory mechanism behind this specificity is still elusive. Secreted exosomes can interact with the recipient cell via receptor–ligand interaction, lipid-raft, claveolae, receptor, and clathrin-mediated endocytosis, micropinocytosis, and phagocytosis [66]. The mode of uptake depends on the tissue/cell microenvironment (particularly the actin cytoskeleton) and the nature of the cargos but not on the storage conditions. The exosome–cell interaction not only influences the tumor microenvironment but also determines the therapeutic success.

Therapeutic incorporation of bioactive molecules (coding or ncRNA, DNA, antibodies, recombinant proteins, nano-formulations of drugs, and synthetic small molecules) can be performed in two ways. It may be either by direct loading of the isolated/engineered exosomes without involving its biogenesis or by indirect loading, which involves manipulation of the producer cells followed by isolation of the desired exosomes [67]. 

#### 4.2.1. Simple Incubation

It is the incubation of exosomes with a high amount of hydrophobic target molecules in a single solution to promote concentration gradient-dependent diffusion with gentle shaking. It is often coupled with density gradient centrifugation and is mostly used for experimental purposes [68].

#### 4.2.2. Electroporation

Electroporation uses a fine electric pulse to create pores on the exosomal membranes, which are the entry points for the therapeutic agents. This simple method holds good clinical acceptance, but concerns such as exosomal disintegrity or excessive aggregation have to be minimized [69].

#### 4.2.3. Saponin Permeabilization

Saponin permeabilization aids exosomal pore formation via saponin, a non-ionic surfactant. This increases the permeability of exosomes for the cargo molecules. Its specialty lies in the preference for hydrophilic molecules over the more common hydrophobic agents. However, its saponin-induced hemolytic toxicity has to be kept balanced [70].

#### 4.2.4. Sonication

Sonication uses an ultra-sonic probe for the internalization of cargoes into the exosomes. However, this process causes substantial deformation of both exosomes and their cargoes. A specialized multi-layered drug encapsulation can be achieved in this method, where both the membrane and the vesicular core may incorporate the agents but it is not an ideal method for nucleotide incorporation [71].

#### 4.2.5. Extrusion

Extrusion involves mixing the cell and target of interests, which are subsequently passed through a finely porous membrane (100 nm pore size) under controlled temperature and mechanical pressure. In this process, the cells becomes vigorously disintegrated into exosomal mimetics containing those cargoes [72].

#### 4.2.6. Freeze–Thaw Cycles

With repeated cycles of freezing at −80 °C to −195 °C followed by immediate thawing at room temperature (25 °C to 37 °C), freeze–thaw cycles ensure sufficient permeabilization of membrane and encapsulation of particles. This method mimics liposome formation. In this process, the problem of exosomal aggregation becomes less effective than sonication or extrusion [73].

#### 4.2.7. Incubation of Donor Cells

The incubation of donor cells is a co-incubation of exosome progenitor cells and the target drug. In this method, the cells incorporate the cargo molecules and eventually release drug-loaded exosomes [74].

#### 4.2.8. Transfection

Transfection is the most commonly practiced method where the cargo (miRNA, small interfering RNA (siRNA), mRNA, or DNA) is inserted inside the donor cell by different vector systems such as a plasmid vector, lentiviral, or adenoviral packaging system. The transfected cell-derived exosomes successfully contain the desired product; moreover, both transient and stable transfections are applied to fulfill distinct purposes [75]. 

#### 4.2.9. Chemical Conjugation

Chemical conjugation is the process where exosomes are directly transfected using commercially available transfection reagents such as lipofectamine (known as exo-fect). However, the use of lipofectamine compromises the purity and loading capacity of exosomes, making this option inefficient [76].

**Table 2 bioengineering-08-00139-t002:** Methods of exosomal bioengineering: different strategies used for bioengineering the exosomes and their advantages and disadvantages.

Type of Strategies	Functional Utility	Advantage	Disadvantage	Reference
Incubation	Incorporate drugs, nucleic acids, proteins, peptides, nanomaterials	Easiest way of cargo loading	Loading efficiency is low, inserted cargoes are difficult to manipulate	[68]
Surfactant treatment	Incorporate proteins, peptides, nanomaterials	High loading efficiency	Damage exosome integrity, inactivate loaded cargo	[68]
Electroporation	Helps to incorporate drugs, nucleic acids, proteins, peptides, nanomaterials	High loading efficiency	Cargo aggregation	[69]
Sonication	Incorporate drugs, proteins, peptides, nanomaterials	High loading efficiency	Damages exosome integrity	[71]
Extrusion	Incorporate drugs	High loading efficiency	Alters the immune status of exosome	[72]
Freeze–thaw	Incorporate proteins, peptides	High loading efficiency	Cargo aggregation, protein inactivation	[73]
Transfection	Incorporate nucleic acids, proteins, peptides	High loading efficiency for nucleic acids, proteins, and peptides; stable	Cargo sorting is difficult to manipulate	[75]

## 5. Bioengineering of Exosomes

As discussed above, the bioengineering of exosomes combined with the anti-tumorigenic response of exosomes provide a great therapeutic approach. Modern-day bioengineering of exosomes includes cargoes such as ncRNAs, recombinant proteins, soluble proteins, chemotherapeutic drugs, as well as natural phytochemicals, which are encapsulated in it and targeted to specific sites (Figure 1). These modulations help in regulating several signaling pathways associated with cancer treatment. Along with oncogenic signaling molecules, the immune tumor microenvironment also plays a major role in tumor management.

### 5.1. ncRNAs

ncRNAs are a group of RNAs that do not code for proteins. These include miRNAs, lncRNAs, siRNAs, etc. These ncRNAs are selectively packaged in the exosomes and help in regulating several signals related to carcinogenesis.

#### 5.1.1. lncRNAs

lncRNAs are transcripts with lengths exceeding 200 nucleotides but not translated into proteins. Both lncRNAs and exosomes have been investigated separately as biomarkers and pathophysiological mediators with immense therapeutic potential. Exosome-associated lncRNAs have been known to take part in tissue repair and regeneration [77]. LncRNAs that are selectively packed into exosomes modulate tumor growth, metastasis, angiogenesis, and chemoresistance, which in turn regulate cancer progression. The majority of exosomes serve as a natural carrier for lncRNAs, and therefore, lncRNAs used for bioengineering of exosomes need to be selected efficiently [78]. LncRNAs have both tumor-inhibiting as well as tumor-enhancing properties. Exosomes need to be adapted to deliver tumor-suppressive lncRNAs. However, along with tumor suppressive activity, exosomal lncRNAs may also increase the sensitivity of cancer cells to drugs [78]. However, there are very few reports on the artificial transfection of lncRNAs into exosomes. The main challenge for using lncRNAs in the therapy of cancer lies in the fact that circulating lncRNAs need to be protected from nucleases to enable the efficacy of lncRNAs [79]. Loading of lncRNAs by electroporation or sonoporation into exosomes is not feasible due to the unavailability of synthetic lncRNAs [77]. In the absence of synthetic lncRNAs, the use of natural lncRNAs with exosomes as the vehicles is an area of high interest [77]. The collection of exosomes from those cell types with a larger reservoir of lncRNAs, e.g., adult stem cells or stromal cells, are of special interest, [80]. Manipulating the expression of lncRNAs or overexpressing them artificially in certain cell types may stoichiometrically favor the loading of these lncRNAs in the exosomes.

Several lncRNAs which have the potential to be used for therapeutics and can be delivered by exosomes to target sites include LOC285194 which suppressed tumor growth in NSCLC by regulating p53 [81] and FOXF1 Adjacent Noncoding Developmental Regulatory RNA (FENDRR) which too suppressed tumor growth, invasion and migration properties of NSCLC [82]. When exosomes carrying lncRNA MEG3 were delivered to advanced NSCLC cells, the sensitivity of these cells increased towards paclitaxel which decreased proliferation and increased p53 expression [83]. Similarly, lncRNAs MEG3 and nuclear factor kappa light chain enhancer of activated B cell (NF-κB) interacting long noncoding RNA (NKILA) delivered to breast cancer cells induced tumor suppressor activity by inducing p53 and NF-κB signaling pathways [84]. Delivery of lncRNA eosinophil granule ontogeny transcript (EGOT) increased the sensitivity of these cells to paclitaxel due to the upregulation of Inositol 1,4,5-trisphosphate receptor type 1 [85]. Delivery of lncRNAs steroid receptor RNA activator 1 in osteosarcoma cells inhibited proliferation, migration and invasion by sponging of miR-208 [86]. Delivery of lncRNA LINC00520 in cutaneous squamous cell carcinoma inhibited phosphoinositide 3-kinases/ protein kinase B signaling pathway by targeting the EGFR inhibition which in turn suppressed tumor growth, proliferation and migration [87].

Hence, naturally occurring lncRNAs packaged in exosomes may be utilized as a probable therapeutic molecule against cancers in order to deliver site-specific activity.

#### 5.1.2. miRNAs

miRNAs are known to influence several genes regulating carcinogenesis. However, packaging of these miRNAs in the exosomes may lead to their efficient delivery to the target sites and may enhance the production of these miRNAs at the target sites. Thus, miRNAs packaged in exosomes have worked as an efficient therapeutic agent with anti-tumor properties [80]. Synthetically produced miRNAs can be packaged in exosomes and targeted to various sites, where they act as efficient molecules in cancer therapy. These exosomes not only deliver the miRNAs to the target sites but also protect them so that they remain intact and fully functional until they reach their destined targets. After their delivery, miRNAs either silence the translating machinery or degrade the RNA of interest to prevent further translation into proteins [88].

Bioengineered exosomes with a transmembrane domain fused with the GE11 peptide delivered the let-7a miRNA to EGFR-expressing xenograft breast cancer tissue in immunodeficient mice, leading to an anti-tumor effect [80]. Similarly, exosomes carrying miR-146b transfected to marrow stromal cells in male Fischer rats significantly reduced glioma [89]. Exosomes engineered with miRNA-26a targeted HCC and suppressed tumor cell proliferation and migration [90]. Exosomes delivering miR-497 in A549 cells suppressed tumor growth and inhibited the expression of several associated genes such as yes-associated protein 1, hepatoma-derived growth factor, cyclin E1, and vascular endothelial growth factor-A (VEGF-A). Similarly, its delivery to HUVECs drastically reduced angiogenesis by inhibiting VEGF-A [51]. Several other exosomal bioengineering included transfection of miR-143 in THP-1 macrophages of mice, leading to increased expression of that particular miR-143 in tumor, kidneys, and serum of the transfected mice, which showed anti-tumor effect by suppressing tumor growth [91]. 

Exosomal engineering may also enhance the cellular sensitivity to drug response. Exosomes containing miRNA-134 targeting triple-negative breast cancer (Hs578T cells) decreased the expression of Hsp90, which in turn decreased cell proliferation and increased the therapeutic efficacy of anti-Hsp90 treatments in the cells [92]. Exosomes containing miR-122 increased the sensitivity of HCC to sorafenib, leading to decreased tumor size in BALB/c nude mice and thus leading to increased response towards chemotherapy [93]. Exosomes bioengineered with 5-fluorouracil and anti-miRNA-21 targeting colorectal cancer reversed chemoresistance and improved treatment efficiency [94]. Exosomes containing miRNA-Let7a targeting nucleolin-positive cancer cells, specifically leukemic cells, have enhanced the delivery of small RNAs to the targeted tumor sites [95]. miR-221-3p, another miRNA can be manipulated with the help of extracellular vesicle bioengineering, which may be used as a novel therapeutic approach in cancer treatment [96]. miR-221-3p has been known to be partially oncogenic where it escaped VEGF receptor2 (VEGFR2) inhibition, therefore, promoting angiogenesis. However, certain prostate cancer patients have been shown to have low levels of miR-221-3p, showing a dual activity of this particular miRNA [97].

Hence, it may be indicated that, due to the varied anti-tumor effects of miRNA, such as the inhibition of cell proliferation, migration, invasion, and promotion of chemosensitivity, miRNA may be largely exploited in cancer therapy with exosomes as their delivery vehicles.

#### 5.1.3. siRNAs

siRNAs, also known as short interfering RNAs, are double-stranded ncRNAs with 20–27 base pairs in length and that function in the RNA interference network. Exosomes bioengineered with these siRNAs targeting numerous tumorous growths caused RNA interference (RNAi) as well as regulated several genes related to carcinogenesis. Exosomes encapsulated with siRNA by electroporation, target various sites. Arginylglycylaspartic acid exosomes containing KRAS siRNAs delivered to A549 tumors in vivo resulted in KRAS knockdown and subsequent tumor suppression [98]. Similarly, tLyp-1 exosomes bioengineered with SOX2 siRNA delivered to NSCLC reduced proliferation and growth and may be potentially used for cancer therapy [99]. Exosomes engineered with BCR-ABL siRNA inhibited cancer cells and tumor growth in chronic myelogenous leukemic cells [100]. Engineered exosomes with Tpd50 siRNA targeted HER-2 positive cells breast cancer cells and enhanced RNAi therapy [95]. Exosomes containing survivin siRNA (siRNA inhibiting survivin) directed at breast cancer cells enhanced tumor-targeted RNAi [101]. Exosomes bioengineered with VEGF siRNA targeted nucleolin-positive cancer cells (MDA-MA-231, leukemic cells), which enhanced the delivery of small RNAs to targeted tumor sites, leading to an anti-cancer effect [95]. The delivery of siRNA mediated by exosomes to MCF-7 cells targeted cyclin-dependent kinase 4 (CDK4) and arrested the cell cycle, specifically its G1 phase. The exosomal delivery of siRNA to murine xenograft models reduced MCF-7 tumor growth. However, the delivery of these siRNAs to the targets induced no cellular or systemic toxicity, which indicated their potential efficacy in future anticancer therapy [102].

The exosomal cargoes and their delivery to targeted regions along with the detailed mechanisms have been tabulated below (Table 3).

### 5.2. Bioengineered Exosome-Based Immune Modulation

Cancer immunotherapy is regarded as a promising tool to fight against cancer. However, the major complexity in the success and failure of cancer immunotherapy is the presence of an immunosuppressive tumor microenvironment. Exosomes mediating intercellular communication affect immunotherapeutic effectiveness. Bioengineered exosomes are involved in inducing immune responses in the tumor microenvironment [103] (Figure 2). A 3D microfluidic cell culture platform has been developed for the production of intact major histocompatibility complex (MHC) peptide surface-engineered exosomes. The system provides an automated, real-time production of exosomes for therapeutic purposes. The harvested exosomes exhibited the ability of antigen presentation and activation of T cells by the surface expression of a melanoma peptide. Therefore, tumor antigen-specific T cell activation was induced by the engineered exosomes compared with the non-engineered exosomes [104]. Along with lymphocytes, DC-based exosomes in cancer immunotherapy are beneficial because they consist of abundant surface lactadherin, which is needed for exosome uptake. Exosomes are also useful in the development of vaccines due to their presence of surface-bound proteins, which originated from their progenitor cell membrane [5].

#### 5.2.1. Lymphocytes

Plasma exosomes derived from human peripheral blood may be used as a successful delivery system for siRNA to human blood monocytes and lymphocytes, causing specific silencing of mitogen-activated protein kinase 1 [105]. Invariant natural killer T (NKT) cells are a type of cell that shares both innate and adaptive immune cell characteristics and were found to have an important anticancer response. NK cells exhibit rapid immunity against malignancies. Exosomes derived from NK cells also exhibit anti-tumor effects in melanoma [106]. Once activated, iNKT cells secrete interferon-γ (IFN-γ) and IL4, which exert their impact on NK, B, and T cell immune responses. Alpha-galactosyl ceramide (αGC) is a glycolipid that was found to upregulate the activation of iNKT cells in vivo but the injection of soluble αGC anergizes the iNKT cells. However, exosomes loaded with ovalbumin and αGC may induce the activation of iNKT cells by overcoming the anergic condition and subsequent amplification of specific anti-tumor adaptive immune responses both in vitro and in vivo. This bioengineered exosome induced NK and γδ T-cell innate immune responses, induced ovalbumin specific B and T cell immune responses, and reduced tumor growth in ovalbumin expressing melanoma in a mouse model [107]. Myeloma-derived exosomes engineered with membrane-bound Hsp70 efficiently stimulated type 1 Th1 cell responses, CD8+ cytotoxic T cell responses, and maturation of DCs. Therefore, these Hsp70 engineered exosomes may represent an effective exosome-based vaccine [108]. 

Recently, genetically engineered T cells expressing chimeric antigen receptors (CAR-T cells) are emerging as a promising immunotherapeutic anti-cancer treatment strategy. A combination of exosomes and CAR-T cells is expected to have induced anti-tumor responses. Exosomes secreted from CAR-T cells carry CAR on their surface. These CAR exosomes inhibit tumor growth and express greater cytotoxic molecules both in vitro and in vivo. Additionally, unlike CAR-T cells, CAR exosomes do not express programmed cell death protein 1, remain unaffected by programmed cell death ligand 1 treatment, and exhibit better anti-tumor properties [109]. Another engineered exosome is synthetic multivalent antibodies retargeted (SMART) exosomes. Exosomes genetically engineered to display both anti-human HER2 antibodies and anti-human CD3 result in the formation of SMART exosomes. This exosome can target both human EGFR 2 of breast cancer cells and CD3 T cells. The exosome smartly redirects the activated T cells towards HER2-expressing breast cancer cells and exhibits a potent anti-tumor response. This SMART exosome might provide a promising platform in the development of next-generation immune-nanomedicines [110].

#### 5.2.2. Dendritic Cells (DC)

Large quantities of exosomes are released by DCs. These exosomes transfer antigen-loaded MHC class I and II molecules to other DCs, leading to the induction of immune response [111]. Exosomes derived from DCs are also capable of inducing T cell immune responses by decorating functional surface MHC/peptide complexes. A phase I clinical trial of vaccination with autologous DC-derived exosomes in stage III/IV metastatic melanoma patients have highlighted the safety of the administration of exosomes. However, melanoma antigen gene (MAGE)-specific T cells were not generated by the DC-derived exosome vaccine but enhanced the effector function of NK cells in the peripheral blood of melanoma patients [112]. Another phase I clinical trial with autologous DC-derived exosomes loaded with MAGE tumor antigens showed a stable long-term prognosis of the disease and activation of immune cells in NSCLC patients. MAGE-specific response of T cells and lytic activity of NK cells were induced by the DC-derived exosomes in lung cancer patients [113]. Another phase II clinical trial, dendritic cell-derived exosomes pulsed with MART1, MAGE tumor antigen, boosted the anti-tumor response of NK cells in unresectable NSCLC patients (NCT01159288) [114]. Thus, clinical studies suggested that DC-derived exosome vaccination may induce an innate and adaptive immune response in cancer patients and can be administered safely. On the other hand, melanoma TEXs were used in DC-based immunotherapy. Here, DCs loaded with TEXs showed increased overall survival compared with DCs loaded with tumor lysate in tumor-bearing BALB/c mice [115]. The α-fetoprotein (AFP)-expressing DC-derived exosomes elicited potent antigen-specific immune responses and significant suppression of HCC tumor growth and prolonged survival rates in mice. Therefore, AFP-enriched DC-derived exosomes may provide an option for cell-free vaccine-mediated immunotherapy [116]. DC-derived exosomes harboring functional MHC/peptide complexes promoted NKG2D-dependent activation of NK cells and exerted non-MHC-restricted anti-tumor response [117]. By using pulsed-peptides, DC-derived exosomes may be further studied for anti-cancer treatments. Pancreatic TEX-loaded DCs significantly prolonged the survival time in C57BL6 mice. However, combined exposure of cytotoxic drug (sunitinib, ATRA, and gemcitabine) treatment and DC-TEX vaccination resulted in induced T cell activation in the tumor, reduced myeloid derived suppressor cells, and increased survivability of tumorigenic mice [118].

#### 5.2.3. Macrophages

Exosomes derived from M1 macrophages translocate towards lymph nodes after subcutaneous injection. These M1 exosomes are taken up by the DCs and macrophages, which in turn induce the secretion of Th1 cytokines. M1 exosomes upregulated the lipid calcium phosphate (LCP) nanoparticle-encapsulated Trp2 vaccine activity and induced antigen-specific T cell response. The study showed that exosomes derived from M1 macrophages acted as a potent immunopotentiator (better than CpG oligonucleotide) in the growth inhibition of melanoma when used with the LCP nanoparticle vaccine. Thus, M1 exosomes may be used as a potent vaccine adjuvant [119]. Another study showed the potential of exosomal CpG oligonucleotides in murine melanoma. Genetically engineered streptavidin-lactadherin-expressing exosomes (SAV exosomes) were combined with biotinylated CpG DNA to form a CpG-SAV exosome. This modified exosome effectively activated DCs with enhanced tumor antigen presentation. Therefore, immunization with CpG-SAV exosome is an effective anti-tumor immunotherapy [120]. Both CpG exosomes and LCP nanoparticle exosomes might be used as an important anti-cancer exosome-based vaccine.

#### 5.2.4. Indirect Bioengineering of Exosomes for Immune Modulation

Not all exosomes are directly engineered for anti-tumor response. In some cases, exosomes isolated from engineered cells/treated cells may also regulate immune responses. Histone deacetylase inhibitors such as MS-275, commonly used as an epigenetic drug, modulate the exosome secretion coated with increased Hsp70 and MHC class I chain-related protein B expression. This MS-275-mediated modification of exosomes significantly induced NK cytotoxicity and proliferation of peripheral blood mononuclear cells [121]. CD40 signaling is essential for DC activation. In a study, exosomes isolated from CD40L gene-modified Lewis lung tumor cells were found to induce the maturation of DCs and IL-12 secretions. These CD40L exosome-treated DCs induce greater proliferation of tumor antigen-specific T cells and may be used as an effective vaccine [122]. Therefore, modifications of donor cells of exosomes may exert a significant anti-tumor response. 

Melphalan (a genotoxic agent that produces genotoxic stress) is commonly used in the clinical management of multiple myeloma patients. Melphalan induced the release of exosomes from multiple myeloma cells. These myeloma-derived exosomes stimulated NK cell-mediated IFN-γ production but did not affect NK cell cytotoxic activity in an HSP70/Toll-like receptor (TLR2)/NF-kB dependent pathway. Hsp70+ exosomes are also found in the bone marrow of multiple myeloma patients, which may exert immunomodulatory effects. Therefore, a chemotherapeutic drug may induce innate immune responses by stimulating the release of exosomes carrying damage-associated molecular patterns such as Hsp70 [123]. 

### 5.3. Chemotherapy

Designing biomimetic nano-formulations without disturbing the structural and functional integrity of the therapeutic molecule has become a primary challenge in high throughput cancer chemotherapy (Table 4). Exosomes are a nano-sized extracellular messenger vesicle suitable for tissue-specific therapeutic drug delivery [124]. Due to their biological uniqueness, exosomes have superior organ enrichment, an in-built homing capacity, cancer cell-specific uptake, and a sustained release ability compared with readily available synthetic nano-drug carriers such as liposomes, micelles, and nanogels. Moreover, nano-toxicity and rapid drug clearance by the body’s immune system, which were associated with previous technologies, are missing in this exosomal delivery system by virtue of their natural origin [125]. The higher secretory ability of the TEX in comparison with their normal counterparts makes them suitable for non-toxic and non-immunogenic drug delivery vehicles for different types of cancer models. Moreover, exosomes possess the unique property of equal affinity for both hydrophilic and hydrophobic chemotherapeutic agents, and they are capable of bypassing immune surveillance and crossing the BBB [124].

Exosomes extracted from RAW 264.7 macrophages were loaded with paclitaxel by three high-capacity methods (incubation, electroporation, and sonication) and were used for treating the drug-resistant and sensitive sub-lines of MDCK renal carcinoma cells. Almost a fifty times increase in cytotoxicity was achieved compared with the free paclitaxel because of the sustained-release property and evasion of drug-efflux machinery by the paclitaxel-incorporated exosomes. These paclitaxel filled exosomes after intranasal administration fully co-localized in the metastatic lesions of murine lung metastases model with high drug-efflux phenotype and showed strong anticancer potential [126]. Milk-derived exosomes packed with paclitaxel or docetaxel have also presented better anti-tumor effects by virtue of their time-dependent slow release of the drugs specifically at the tumor site. The significant anti-inflammatory effect was also observed with this drug–exosome formulation [127]. A recent study has shown that synthetically personalized exosome mimetics (EMs) may serve as an alternative vesicle for the delivery of drugs. Human mesenchymal stem cell-derived EMs mixed with Paclitaxel exhibited decreased tumor growth in vivo and reduced the viability of breast cancer cells in vitro [146].

Exosome-sheathed doxorubicin-loaded porous silicon nanoparticles (DOX@E-PSiNPs) prepared by electroporation exhibited 2.1 fold increased intracellular uptake and retention in a spheroid model of murine hepatocarcinoma H22 cancer stromal cells. DOX@E-PSiNPs downregulated the multi-drug resistance-associated drug efflux pumps, induced cytotoxicity, and reduced frequency and size of H22 cancer stem cells spheroid colonies. Improved cross-reactive cytotoxicity was also obtained with this DOX@E-PSiNPs treatment in murine melanoma B16-F10 cells. In the H22-xenograft murine hepatocarcinoma tumor model, DOX@E-PSiNPs caused a higher tumor accumulation than non-cancerous neighboring cells and deeper penetration of the drug doxorubicin [128].

In another study, a cocktail of chemotherapeutic drugs—doxorubicin, 5-fluorouracil, gemcitabine, and carboplatin—were packaged in macrophage (U937 or Raw264.7)-derived exosomes by incubating the cells with these drugs. The drug-loaded biomimetics of exosomes are capable of in vitro anti-inflammatory endothelial cell death. Equivalent in vivo tumor targeting and tumor growth retardation without nonspecific toxicity was also achieved with this loaded exosome-mimetics in comparison with free drugs [129].

Autologous TEX was incubated with gemcitabine (one of the first choice chemotherapeutic drugs for the treatment of pancreatic cancer) either by simple incubation or by sonication, and these gemcitabine-loaded exosomes (ExoGEM) were reintroduced in pancreatic cell line PANC-1. This ExoGEM presented target-specific sustainable release and better intracellular retention in vitro. In the pancreatic-xenograft model, this exosomal formulation inflicted less immunogenicity, off-target toxicity, better tumor growth-inhibition, and tumor-free survival [130].

A2780, a human ovarian cancer cell line when incubated with cisplatin (one of the most-used chemotherapeutic drugs) and then UV-irradiated produced an ample amount of cisplatin integrated-exosomal micro-vesicle. This carrier system retarded the growth of human ovarian tumors in SCID mice and facilitated the survivability of the tumor-challenged animal in comparison with cisplatin alone [131].

### 5.4. Exosomal Delivery of Small Molecules

The main target of cancer research is to develop improved anticancer strategies, which can precisely target cancer cells, causing no or less damage to healthy normal cells. In this context, the usefulness of bioactive phytoagents may be promising because of their easy accessibility, selective cancer killing, minimal side effects, and multimodal functionality [147]. However, along with all of these great benefits, they have some practical limitations too such as poor bioavailability due to insolubility or incomplete penetration, nonspecificity, low therapeutic index, rapid biotransformation, and elimination. To overcome such challenges, a micro-level targeted delivery system such as exosomal carriers may be a resourceful alternative to fully utilize the antineoplastic potential of these natural small molecules [125]. Natural/synthetic/semi-synthetic small molecules may be loaded into exosomes by both direct (during biogenesis) and indirect (manipulation of the producer cells) methods. Plenty of experimental pieces of evidence strengthen the application of exosomes as the carrier of cancer-curative phytochemicals.

#### 5.4.1. Natural Phytochemicals

Flavonoids (e.g., myricetin, quercetin, and kaempferol) and soya saponins from black bean extracts are excellent anticancer agents as they can reduce the oxidative stress-induced cancer risk and induce apoptotic toxicity in cancer cells. TEXs isolated from various human cancer cells of different origins—mammary (MCF7), prostate (PC3), colon (Caco2), and liver (HepG2)—were electroporated with black bean-derived phytochemicals. When cancer cells were inserted with modified TEXs, they showed higher accumulation of the phytochemicals, which in turn caused apoptosis and cell cycle arrest [132]. 

When the cow milk-derived exosomes were simply incubated with berry-derived anthocyanidin (anti-oxidant, anti-inflammatory, and anti-proliferative phyto-compound), a heightened anti-tumor efficacy was observed [133]. Along with this profound anti-inflammatory effect, reversal of drug resistance in cancer cells and selective low-toxicity in normal counterparts was also observed in cancers of the lung, prostate, ovary, breast, pancreas, etc. and in vivo xenograft models [134].

Curcumin, the most bio-active polyphenol from turmeric, presented a five-fold higher concentration and almost four-fold higher stability than free curcumin when packaged with EL-4 (murine lymphoma) cell-derived exosomes via mixing and gradient centrifugation. These curcumin-filled exosomes (Exo-Cur) showed almost five- to ten-fold higher curcumin content for a longer period in peripheral blood upon oral administration when studied in murine-xenograft model. As a result, a heightened anti-inflammatory and anti-cancer effect was also obtained with Exo-Cur in different cancer cell lines or tissues such as the breast, lung, and cervix [148]. In another study, the same Exo-Cur markedly retarded the tumor growth of GL26-xenograft murine brain tumor model [141]. 

Chemopreventive phytochemicals such as withaferin A or anthocyanidins were packaged within cow milk-derived exosome via mixing and centrifugation. They showed significant toxicity in lung cancer (A549 and H1299) cells and in breast cancer (MDA-MB-231 and T47D) cells, as evidenced from a much-reduced IC_50_ value of the encapsulated from than the free form of those chemopreventive agents. This exosomal formulation has even minimized NF-κB-mediated inflammatory stress. However, all of these anti-cancer effects of loaded exosomes are dose-time dependent and highly cancer-specific, leaving the normal healthy cells (bronchial BEAS-2B) unaffected. The A549-xenografted animal model has also shown tumor growth retardation and volume-shrinkage upon oral treatment of the abovementioned exosomal formulation [127].

Honokiol, an anti-tumor phytochemical from magnolia when packed in MSC-derived exosomes by sonication proved to be more beneficial than the free compound in various cancer cell lines such as pancreatic (MiaPaCa and Colo357), breast (MDA-MB-231), ovarian (SK-OV-3), colon (HT-29), and prostate (LNCaP) cells. Improved therapeutic potential in terms of the upregulation of cell-cycle arrest and apoptotic response, and the downregulation of survival-associated factors and clonogenic properties was achieved owing to the better cellular concentration of honokiol in exosome-encapsulated cases over the administration of free honokiol [135].

Celastrol, a triterpenoid phytochemical packaged in milk-derived exosome caused a significant dose-time-dependent growth inhibition when compared with celastrol alone in NSCLC (A549 and H1299) cell lines by decreasing NF-κB-mediated inflammation and by increasing endoplasmic reticulum-stress mediated apoptosis. The superior anti-tumor effect of this celastrol-loaded exosome was also proved in the lung cancer xenograft model, where no unwanted systemic toxicity was found to be an added advantage of this exosome formulation than the nonspecific free celastrol [140].

#### 5.4.2. Other Small Molecules

Porphyrine, a photo-sensitive synthetic drug, showed remarkable cellular retention compared with the only drug or free exosome when integrated with MDA-MB-231-derived TEX via various methods such as passive mixing or active electroporation/saponin-assisted incubation/extrusion/dialysis. On reintroduction into that breast cancer cell line, it resulted in significant cancer cytotoxicity in presence of light [139]. 4T1-derived TEX was co-incubated with sinoporphyrin sodium to form a nano-sized ultrasonic sound sensitizer, which had both therapeutic and imaging properties. This functionalized exosomal formulation showed promising therapeutic efficacy. On one hand, they induced ROS-driven death-signaling and inhibited metastasis, while on other hand, they facilitated simultaneous tumor-imaging [136].

A HeLa-derived exosome that acts as a multifunctional drug carrier may be stably incorporated with more than one photo-therapeutic drug such as porphyrin, indocyanine, etc. from a mixture. These anti-tumor components of the multifunctional exosome upon photo-irradiation worked simultaneously and synergistically for successful cancer treatment in a human lymphoblastic CCRM-CEF xenografted murine model [149].

Aspirin, an excellent cardio-protective non-steroidal anti-inflammatory drug and an emerging anti-cancer agent, with the help of breast (MDA-MB-231) and colorectal (HT29) TEX was efficiently delivered back to those cancer cells with a greater cellular accumulation of aspirin than its free form. This aspirin-loaded exosome showed increased cancer toxicity in terms of more apoptotic and autophagic cell death in both in vitro and in vivo systems. A novel cancer stem cell eradication by this exosomal-aspirin was also observed [137].

JSI124, a signal transducer and activator of transcription3 inhibitor cum anti-proliferative agent when packaged in TEX (Exo-JSI124), introduced apoptotic cytotoxicity in GL26 murine glioma and showed an anti-inflammatory effect in this microglia-xenografted animal model after nasal administration of JSI124-encapsulated exosome [132].

By the virtue of its BBB-crossing ability, serum exosomes may efficiently deliver therapeutic agents such as dopamine, a catecholamine neurotransmitter, or catalase, an anti-oxidant enzyme, to murine brain-degeneracy models from a mixture after preserving their complete functionality [63].

Exosomes can successfully express a biotin-streptavidin-fused luciferase by lentiviral transfection, compatible with fluorescence or chemiluminescence-guided tracking [150]. Fluorophore-conjugated antibodies against exosomal markers produced by coincubation are another means of in vivo tracking of exosomes [151]. These technical advancements have enabled exosomes to be used as a real-time imageable device to study its distribution, penetration, biological half-life, etc. 

Tissue MSC-derived exosomes were successfully loaded with venofer, a Fe_3_O_4_-labelled nanoparticle by incubation of the MSCs with venofer. This iron-loaded MSC exosome inhibited the proliferation rate of prostate cancer (PC3) cells in a dose-dependent manner. After successful incorporation in the tumor site, these magnetic exosomes resulted in target-specific tumor ablation. This antitumor effect of these loaded exosomes was further increased with magnetic hyperthermia [138].

Serum reticulocyte-derived exosomes were used to design a stable yet functionalized super-paramagnetic Fe_3_O_4_ nanoparticle cluster (SMNC-Exo). This self-assembled exosome-based nano-sized drug carrier may successfully deliver chemotherapeutic drugs (e.g., doxorubicin) in a sustained but targeted manner better than the free drug. A stronger anti-tumor response may be achieved with the aid of an external magnetic field in the subcutaneous model of murine hepatoma [152].

### 5.5. Recombinant Protein

In recent studies, exosomes have been reported to express recombinant proteins that could be used as vaccine strategies or means of drug delivery in cancers. For example, carcinoembryonic antigen and HER2 were coupled to the CIC2 domain of lactadherin. This fusion protein enhanced the immunogenicity of different human tumor-associated antigens and augmented the antitumor effect both in vivo and in vitro [153]. A bio-engineered exosome with a native soluble fragment of human hyaluronidase (PH20 and Exo-PH20) exhibited degradation of hyaluronan in the deep tumor foci. This hyaluronan degradation inhibited tumor growth, augmented T cell infiltration, and elevated drug diffusion into the tumor [142]. More specifically Exo-PH20 was found to activate the maturation and migration of CD103+ DCs that ultimately activated CD8+ cells. Thus, CD8+ T cells and DCs together inhibited tumor growth in vivo [143]. However, the native glycosyl phosphatidyl inositol (GPI) anchored form of hyaluronidase was enzymatically more active than the truncated recombinant form [142]. Human decay-accelerating factor-derived GPI-anchor signal peptide was fused with EGa1 nanobodies to produce a high-affinity ligand for EGFR. This recombinant protein dramatically improved ligand binding to EGFR-expressing cancerous cells [154]. In another study, TNF-α anchored exosomes were coupled with superparamagnetic iron oxide nanoparticles along with cell-penetrating peptides. This fusion protein significantly augmented the binding and interaction between TNF-α and its membrane receptor TNFRI, resulting in TNFRI-mediated apoptosis and repressed tumor growth [144]. Interestingly, engineered exosomes with signal regulatory protein α (SIRPα) were able to put an immune checkpoint blockade to disrupt the CD47-SIRPα interactions on phagocytic cells. As a result, SIRPα exosomes augmented macrophage engulfment, T cell infiltration, and inhibition of tumor growth in vivo [145]. Extracellular vesicle-based delivery of tyrosine kinase inhibitors resulted in the reversion of radioiodine-resistant thyroid cancer cells to radioiodine-sensitive cells [155]. Even human liver stem cell-derived extracellular vesicles increased the sensitivity of cancer stem cells towards tyrosine kinase inhibitors [156]. Extracellular vesicles mediated transport of sodium iodide symporter enhanced radioiodine uptake in hepatocellular carcinoma [157]. Though exosome trafficking, function, and stability are not very well understood to date, this nature-based vehicle of protein cargo may be implemented for exosome-mediated therapeutics.

### 5.6. Fusogenic Exosome

Yang et al. have developed a fusogenic exosome that is a well-designed recombinant exosome harboring viral fusion-mediated glycoproteins (FMGs). These fusogenic exosomes can fuse with the target cancer cell membrane to deliver FMGs. They modify the target membrane to express viral pathogen-associated molecular patterns (PAMPs) that can be recognized by the immune cells as ‘non-self’ and can exert an anti-tumor effect [158]. Several studies showed that exposure to PAMPS by vaccination exerted therapeutic benefits in cancer treatment. The formation of this xenogenized tumor by the expression of viral PAMPs induced their recognition and phagocytic engulfment by DCs and potent anti-tumor immune response. A combination of fusogenic exosomes and anti-programmed death ligand-1 treatment effectively expressed anti-tumorigenic responses [159]. However, applications of such fusogenic exosomes need further investigations.

### 5.7. Vexosomes (Vector Exosomes)

Apart from RNAs, chemotherapeutic drugs, and other molecule-mediated engineering, another type of exosome modification is the formation of vexosomes. Maguire et al. have termed vexosomes as vector exosomes that involve viral packaging of exosomes. Adeno-associated virus (AAV) vectors exhibited efficient drug delivery both in vitro and in vivo. During the production of AAV vectors, a fraction of the vectors that remained associated with the exosomes were termed as vexosomes, and these showed high transduction efficacy. Therefore, vexosomes might be a promising strategy for gene delivery into tissue [160]. Exosomes containing AAV capsids were used to deliver DNA to human glioblastoma cells [160]. In another study, Khan et al. developed AAV serotype 6 vexosomes containing an inducible caspase 9 (iCasp9) suicide gene. This modified AAV-iCAsp9 vexosomes along with a pro-drug (AP20187) caused a significant reduction in cell viability in HCC cells [161]. Studies with vexosomes are very few, which warrant more elaborate studies to obtain an efficient drug delivery system.

## 6. Future Prospects and Conclusions

Recent advancements in the engineering of exosomes have increased the curiosity of researchers for developing more advanced and novel therapeutic approaches. Several companies were found to manufacture bioengineered exosomes for therapeutic applications. Despite several developments, problems associated with large-scale production and the purification of exosomes need to be addressed more precisely. Highly advanced, less time-consuming and high-production yield methods might be associated with future therapeutic and diagnostic platforms. Natural exosomes have several potentials, but clinically, they are associated with several limitations. To overcome the limitations of natural exosomes, designer exosomes were developed using parental cell-based engineering for targeted delivery of drug and functional molecules to specific recipient cells [162]. These designer exosomes are also involved in vaccine development [162]. A reliable large-scale isolation method of exosomes and more information on the functional characteristics, biogenesis, and exosomal contents would significantly enlighten new advanced opportunities for using exosomes as anti-cancer therapeutics. Future research on the natural heterogeneity of exosomes needs to be explored for developing exosomal drugs with greater efficacy.

Several decades of study have pointed out significant and promising methods of engineering exosomes with induced anti-cancer potential. In-depth understanding of the properties of engineered exosomes for targeting metastasis might provide a significant therapeutic approach for an increased survival rate in cancer patients. Exosome vaccines provide a promising therapeutic approach. Exosomal modifications with ncRNAs, chemotherapeutic drugs, recombinant proteins, and other small molecules have yielded encouraging anti-tumor responses that may support the future development of clinical practices. To utilize this nanoscale drug delivery platform of exosomes, integrated use of new technologies and basic research will set the foundations for their clinical acceptance.

## Figures and Tables

**Figure 1 bioengineering-08-00139-f001:**
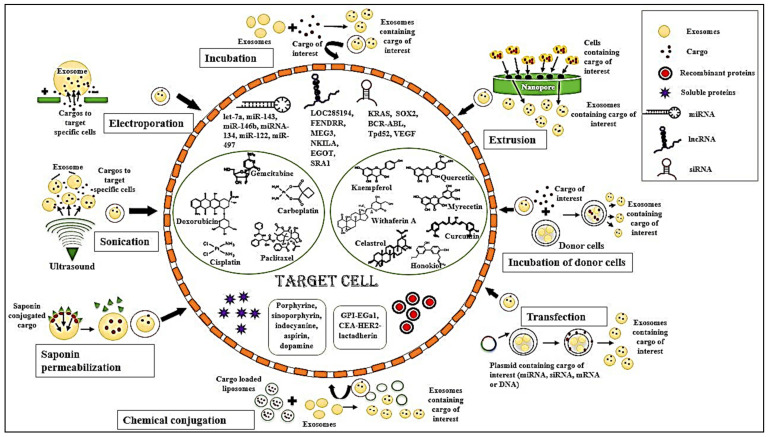
Bioengineering of exosomes: Different methods of cargo incorporation into exosomes and different types of cargo internalized in the exosomes targeted to various target cells.

**Figure 2 bioengineering-08-00139-f002:**
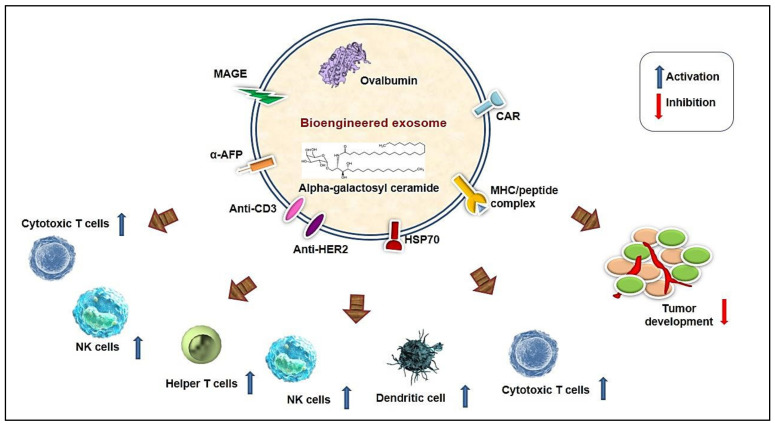
Bioengineering of exosomes for immune regulation: Modified exosomal cargoes and other molecules surface molecules regulate the activation of immune response and the inhibition of tumor development. CAR, chimeric antigen receptor; HER2, human epidermal growth factor receptor 2; HSP, heat shock proteins; MAGE, melanoma antigen gene; NK, natural killer cell.

**Table 3 bioengineering-08-00139-t003:** Noncoding RNAs as exosomal cargoes for cancer prevention and therapy: noncoding RNAs such as lncRNA, miRNA, and siRNA as encapsulated cargoes, used for cancer prevention strategies.

Nature of Cargo	Encapsulated Cargo	Nature of Study	Model	Target Tissue Type	Function	Mechanism	Reference
lncRNA	LOC285194	In vitro	A549 and H1299	NSCLC	↓Tumor growth	↑p53	[81]
	FENDRR		H1650, HCC827, H1975, and A549	NSCLC	↓Tumor growth, ↓ Invasion, ↓migration, and ↑apoptosis	↓miR-761	[82]
	MEG3		A549	Advanced NSCLC	↑Sensitivity to paclitaxel and ↓proliferation	↑p53	[83]
	MEG3 and NKILA		MCF-7 and BT474	Breast cancer	↑Tumor suppression	↑p53 and ↑NF-κB signaling pathways	[84]
	EGOT		BT549	Breast cancer	↑Sensitivity to paclitaxel	↑ITPR1, ↓GLI1, ↓smoothened protein, ↓protein patched homolog 1, and ↓ HHIP	[85]
	SRA1		SAOS-2, MG63, U2OS, SJSA1, and human osteoblast hFOB	Osteosarcoma	↓Proliferation, ↓migration, and ↓invasion	Sponging of miR-208	[86]
	LINC00520		A431 cSCC (cutaneous squamous cell carcinoma)	Cutaneous squamous cell carcinoma	↓Tumor growth, ↓proliferation, and ↓migration	↓PI3K/Akt and ↑EGFR	[87]
miRNA	miR-497	In vitro	A549	Lung cancer	↓Tumor growth	↓YAP1, ↓HDGF, ↓CCNE1, and ↓VEGF-A	[51]
			Coculture of A549 and HUVECs	Human umbilical vein endothelial cells (HUVECs)	↓Angiogenesis	↓ VEGF-A	[51]
	miRNA-26a		Class B type 1-expressing liver cancer cells	Hepatocellular carcinoma (HepG2)	↓Tumor cell proliferation and ↓migration	-----	[90]
	miR-143		THP-1 macrophages	Metastasis-associated in colon cancer-1 (MACC1)	↓Cell growth, ↓migration, and ↓invasion	↓EGFR and ↓NF-κB	[91]
	Let-7a		Hs578Ts(i)8 cells	Triple negative breast cancer (Hs578T cells)	↓Cell proliferation and ↑therapeutic efficacy of anti-Hsp90	↓ Hsp90 expression, ↓STAT5B, and ↓Bcl-2 levels	[92]
	5-fluorouracil anti miRNA-21		HCT-1165FR	Colorectal cancer (HCT-119)	↓Chemoresistance and ↑treatment efficiency	↑PTEN	[94]
	miRNA-Let7a		MDA-MA-231, leukemic cells	Nucleolin-positive cancer cells	Anti-cancer effect	↑Delivery of small RNAs to the targeted tumor sites and ↓EGFR	[95]
	miR-134	In vivo	RAG2–/– mice	EGFR-expressing xenograft breast cancer tissue	Anti-tumor effect	↓K-RAS, ↓H-RAS, and ↓N-RAS	[80]
	miR-146b		Male Fischer rats	Marrow stromal cells	↓Glioma	-----	[89]
	miR-122		BALB/c nude mice	Hepatocellular carcinomas (HCCs)	↑Sensitivity towards sorafenib and ↓tumor size	↓ CCNG1, ↓IGF1R, ↓ADAM10,↑ Caspase 3, and ↑Bax	[93]
siRNA	Tpd52 siRNA	In vitro	HER-2 positive cells (SKBR3 cells)	Breast cancer	↑RNAi therapy	↓TDP52 expression by about 70%	[95]
	VEGF siRNA		MDA-MA-231	Nucleolin-positive leukemic cells	Anti-cancer effect	↑Delivery small RNAs to the targeted tumor sites	[95]
	KRAS siRNAs		A549	Lung cancer	↓Tumor suppression	↓ KRAS expression	[98]
	SOX2 siRNA		NSCLC	Lung cancer	↓Proliferation and growth	-----	[99]
	BCR-ABL siRNA		LAMA84, K562, and K562R	Chronic myelogenous leukemia	↓Cancer cell growth and ↓size of tumors	↓BCR-ABL expression by 17% to 45%	[100]
	Survivin siRNA		MDA-MB-468	Breast cancer cells	↑RNAi therapy	-----	[101]
	siRNA		MCF-7	Breast cancer	Anti-tumor effect	↓CDK4 and cell cycle arrest in G1 phase	[102]

Abbreviations: ADAM10, a disintegrin and metalloproteinase domain-containing protein 10; BCL2, B-cell lymphoma 2; BCR, ABL, breakpoint cluster region protein, tyrosine-protein kinase; CCNG1, cyclin-G1; CDK4, cyclin-dependent kinase 4; EGFR, epidermal growth factor receptor; EGOT, eosinophil granule ontogeny transcript; Gli1, glioma-associated oncogene; HGDF, hepatoma-derived growth factor; HHIP, hedgehog-interacting protein; Hsp90, Heat shock protein 90; IGF1R, insulin-like growth factor 1; ITPR1, inositol 1,4,5-trisphosphate receptor type 1; MEG3, maternally expressed Gene 3; NF-κB, nuclear factor kappa light chain enhancer of activated B cells; NKILA, NF-κB interacting long noncoding RNA; PI3K/Akt, phosphoinositide 3-kinases/protein kinase B; PTEN, phosphatase and tensin homolog; STAT5, signal transducer and activator of transcription 5; VEGF, vascular endothelial growth factor; YAP1, yes-associated protein 1.

**Table 4 bioengineering-08-00139-t004:** Exosomal bioengineering for cancer diagnosis and therapeutics.

Source of Exosomes	Encapsulated Cargo	Target Cancer Model	Loading Method	Tumorigenic Effect	Mechanism	Reference
**Chemotherapeutic Drugs**						
**In vitro**						
RAW 264.7 macrophage	Paclitaxel	Renal carcinoma (MDCK) cells	Incubation, electroporation, and sonication	↑Cytotoxicity, ↓drug-efflux pump, and resistance reversal	↓Pgp	[126]
Milk from pasture-fed Holstein and Jersey cows	Paclitaxel and docetaxel	A549, H1299, MB-231, and T47D	Incubation and centrifugation	Anti-tumor effect and ↑anti-inflammatory effect	___	[127]
H22, Bel7402, or B16-F10 cells	Doxorubicin	H22 and B16-F10 cells	Electroporation	↑Cytotoxicity,↑tissue-enrichment, ↓spheroid size, and ↓nonspecific adversities	___	[128]
U937 or Raw264.7 macrophages	Doxorubicin, 5-fluorouracil, gemcitabine, and carboplatin	HUVEC	Incubation and sonication	↑Anti-inflammatory, ↓nonspecific adversities, and ↓tumor growth	___	[129]
PANC-1 cells	Gemcitabine	PANC-1 cells	Incubation or sonication	↓Nonspecific adversities and ↓tumor growth	___	[130]
H22 and A2780 cells	Cisplatin	H22 and A2780 cells	Incubation and UV-irradiation	↑Cytotoxicity and↓ drug efflux	___	[131]
**In vivo**						
H22 and A2780 cells	Cisplatin	H22 and A2780 cell xenografted BALB/c mice	Incubation and UV-irradiation	↑Tumor growth inhibition and ↑survivability of tumor-challenged mice	___	[131]
**Small Molecules**						
**In vitro**						
Milk from pasture-fed Holstein and Jersey cows	Withaferin A, bilberry-derived anthocyanidins, and curcumin	Human lung (A549 and H1299), breast (MDA-MB-231 and T47D) cancer cells, and normal bronchial epithelial (BEAS-2B) cells	Mixing	↓Inflammatory stress	↓ NF-κB	[127]
Human mammary (MCF7), prostate (PC3), colon (Caco2), and liver (HepG2) cells	Black bean-derived (myricetin, quercetin, kaempferol, and soyasaponins)	MCF7, Caco2, PC3, and HepG2 cells	Electroporation	↑Apoptosis and ↑cell cycle arrest	___	[132]
Raw milk from dozens of mid-lactation, pasteurized Jersey cows	Berry-derived anthocyanidin	A549 and H1299; MDA-MB-231 and MCF7; and pancreatic (PANC1 and Mia PaCa2), prostate (PC3 and DU145), colon (HCT116), and ovarian (OVCA432) cancer cells	Simple mixing	↑anti-proliferative and ↑anti-inflammatory	___	[133]
Mature bovine milk	Anthocyanidins	Cisplatin-sensitive (A2780) and cisplatin-resistant (A2780/CP70, OVCA432, and OVCA433)	Mixing	↑anti-proliferative	___	[134]
Mesenchymal stem cells	Honokiol (extracted from Magnolia plant)	Pancreatic (MiaPaCa and Colo357); MDA-MB-231; and colon (HT-29), prostate (LNCaP), and ovarian (SKOV-3) cancer cells	Sonication	↑Cell-cycle arrest, ↑apoptosis, and ↓survival- associated factors	___	[135]
4T1 cells	Sinoporphyrin sodium D	4T1 cells	Incubation	↑Cell death, ↓metastasis, and ↑membrane permeability	↑ROS	[136]
MDA-MB-231 and HT29 cells	Aspirin	MDA-MB-231 and HT29 cells	Freeze–thaw incubation and sonication	↑Cytotoxicity, ↑apoptosis, and ↑autophagy	___	[137]
Mesenchymal stem cells	Venofer	PC3 cells	Incubation	↓Proliferation	___	[138]
MDA-MB-231 cells	Porphyrine	MDA-MB-231 cells	Electroporation/saponin-assisted incubation/extrusion/dialysis	↑Cytotoxicity	___	[139]
**In vivo**						
Mature bovine milk	Anthocyanidins	Ovarian cancer xenografts	Mixing	↓Tumor growth	___	[134]
Milk from pasture-fed Holstein and Jersey cows	Celastrol (a plant-derived triterpenoid)	A549 and H1299 NSCLC xenograft C57BL6 mice	Mixing	↓Tumor growth	___	[140]
EL-4 cells	Curcumin	Lipopolysaccharide -induced brain inflammation in C57BL/6J mice	Mixing, incubation, and centrifugation	↓LPS-induced brain inflammation and ↑apoptosis	___	[141]
EL-4 cells	JSI124	Mouse (H-2b) glioblastoma (GL26) cell-xenografted C57BL/6J mice with brain tumor model	Mixing, incubation, and centrifugation	↓Tumor growth and ↑microglial apoptosis	↓Stat3	[141]
**Recombinant Proteins**						
**In vitro**						
Human embryonic kidney HEK293T cells	PH20	PC3 cells	Transfection	Anti-tumor effect	↓HA	[142]
HEK293T cells	PH20-Oligo-HA	Murine breast cancer (4T1) cells	Transfection	↑Immune response	↓HA, ↑DC cells, ↑IFN-γ, and ↑cytotoxicity of CD8+ T cells	[143]
Human melanoma (A375), breast adenocarcinoma (MCF-7), lung carcinoma (A549), and colon adenocarcinoma (Colo201) cells	CTNF-α exosome SPIONs	A375, MCF-7, A549, and Colo201 cells	Transfection	Anti-cancer effect and ↑apoptosis	↑TNF-α and ↑Cleaved caspase-2,3,8	[144]
HEK293T cells	SIRPα	Human colon adenocarcinoma (HT29) cells, Raji Burkitt’s lymphoma (Raji), and mouse CT26.CL25 colon cancer cells	Transfection	Anti-tumor effect and ↓tumor growth	↓CD47	[145]

Abbreviations: CEA, carcinoembryonic antigen; CTNF-α, cell-penetrating peptides TNF-α; DAF, decay-accelerating factor, ECD, extracellular domain; EGFR, epidermal growth factor receptor; GPI, glycosylphosphatidylinositol; HA, hyaluronic acid; HUVEC, human umbilical vein endothelial cells; IFN-γ, interferon gamma; Pgp, permeability glycoprotein; SIRPα, signal regulatory protein α; SPION, superparamagnetic iron oxide nanoparticles; STAT3, Signal transducer and activator of transcription 3; TNF-α, tumor necrosis factor α.

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
