# Peer review of "Bioengineering of Extracellular Vesicles: Exosome-Based Next-Generation Therapeutic Strategy in Cancer"

_bioengineering, 2021, doi:10.3390/bioengineering8100139_

Round 1

Reviewer 1 Report

Saha et al. briefly review the role of bioengineered of exosomes for next-generation therapeutic strategy in cancer. The review was well written and timely.

  1. I and The International Society for Extracellular Vesicles (ISEV) recommend using the term EVs instead of exosomes and microvesicles. Please change the title from “Bioengineering of exosomes: next-generation therapeutic strategy in cancer” to “Bioengineering of extracellular vesicles: next-generation therapeutic strategy in cancer”.
  2. I would recommend adding more information about the biogenesis of EVs in the section “Structure and composition of exosomes”.
  3. I would recommend adding a dedicated section of internalization of EVs into cells, which is important for therapy.

  1. I would recommend adding flowing papers, which you may consider.

PMID: 33578882, PMID: 30546834, PMID: 30880979, PMID: 30319428.

Author Response

REVIEWER 1:

Comment:

Saha et al. briefly review the role of bioengineered of exosomes for next-generation therapeutic strategy in cancer. The review was well written and timely.

Response:

The authors are thankful for the kind appreciation of the learned reviewer.

Comment:

I and The International Society for Extracellular Vesicles (ISEV) recommend using the term EVs instead of exosomes and microvesicles. Please change the title from “Bioengineering of exosomes: next-generation therapeutic strategy in cancer” to “Bioengineering of extracellular vesicles: next-generation therapeutic strategy in cancer”.

Response:

The authors are thankful for the suggestion. In this review, we have mainly focussed on the exosome-mediated therapeutics. Therefore, the title has been changed into ‘BIOENGINEERING OF EXTRACELLULAR VESICLES: EXOSOME-BASED NEXT-GENERATION THERAPEUTIC STRATEGY IN CANCER’.

Comment:

I would recommend adding more information about the biogenesis of EVs in the section “Structure and composition of exosomes”.

Response:

It is indeed a valuable suggestion. A sub-paragraph on biogenesis of exosomes has been incorporated under heading 2.

Comment:

I would recommend adding a dedicated section of internalization of EVs into cells, which is important for therapy.

Response:

The authors are highly thankful for such minute observation. The first two paragraphs in the section 4.2. Exosomal incorporation methods, stating ‘Natural origin of exosomes render their safety from the body’s immune……tumor microenvironment but also determines the therapeutic success’ has been incorporated as a section of internalization of EVs into the cells. However, the entire section of 4.2 stated the different methods of exosome internalization therapeutically.

Comment:

I would recommend adding flowing papers, which you may consider.

PMID: 33578882, PMID: 30546834, PMID: 30880979, PMID: 30319428

Response:

According to the suggestion PMID: 33578882, PMID: 30546834 and PMID: 30880979 have been incorporated under the section 5.5. Recombinant proteins with reference no. 149, 150 and 151. PMID: 30319428 has been incorporated under the section 5.3. Chemotherapy with reference no. 121.

Reviewer 2 Report

Priyanka Saha et al. reviewed available facts about BIOENGINEERING OF EXOSOMES. The topic is of interest.

Points to be considered:

1) The rationale of why the authors came up with this review.

2) What is the information that is not exactly available that motivated the authors to come up with this information. What are the current caveats and how do the authors highlight the current research in answering them? If not they need to address in future directions.

3) In ncRNA the paradigm of angiogenesis can be expanded: the authors corrected mentioned models pinpointing the relevance of RNA and epigenetics for lung cancers. This reviewer personally misses some important facts regarding other neoplasias and malignancies, such as prostate cancers. Indeed, as an example, the oncogenic function of miR-221-3p as an escape mechanism from VEGFR2 inhibition (refer to PMID: 32131507 and expand).

4) The authors need to highlight what new information the review is providing to enhance the research in progress.

Author Response

Comment:

Priyanka Saha et al. reviewed available facts about BIOENGINEERING OF EXOSOMES. The topic is of interest.

Response:

The authors are thankful for the kind words of encouragement.

Comment:

The rationale of why the authors came up with this review.

Response:

Exosomes being nanovesicles may invade physiological barriers, which were generally impenetrable by other synthetic drug delivery vehicles. The extensive therapeutic benefits of exosomes have made the authors interested with this review. The statements have been incorporated in the first paragraph of introduction section.

Comment:

What is the information that is not exactly available that motivated the authors to come up with this information. What are the current caveats and how do the authors highlight the current research in answering them? If not they need to address in future directions.

Response:

The information has been incorporated in the second paragraph of the introduction section. Most of the previous reviews have focused on the role of exosomes in particular cancer type or only tumor-derived exosomes. However, in our review, we have highlighted the therapeutic strategies till date which may help in improving future pre-clinical and clinical studies. The statements include ‘Dai et al. have reviewed the role…..anti-cancer treatment in all cancer types as well as involving several exosome delivery sources’. The current limitations have been already stated in the first paragraph of section 6. Future prospects and conclusions.

Comment:

In ncRNA the paradigm of angiogenesis can be expanded: the authors corrected mentioned models pinpointing the relevance of RNA and epigenetics for lung cancers. This reviewer personally misses some important facts regarding other neoplasias and malignancies, such as prostate cancers. Indeed, as an example, the oncogenic function of miR-221-3p as an escape mechanism from VEGFR2 inhibition (refer to PMID: 32131507 and expand).

Response:

As per the suggestion the changes have been incorporated as much as possible in the section 5.1.2. miRNAs.

Comment:

The authors need to highlight what new information the review is providing to enhance the research in progress.

Response:

We are extremely indebted to the learned reviewer for this feedback. This review mainly focuses on all the therapeutic strategies of bioengineered exosomes in a single platform, which will allow the researchers to get an idea for future pre-clinical or clinical therapeutics. We have incorporated this information in the second paragraph of introduction: “On the contrary, the present review has tried to provide an insight into the role of exosomes in the regulation of cancer, the strategies of exosomal bioengineering, and their implementation for future anti-cancer treatment against all cancer types. The wide array of exosome delivery modalities, the therapeutic implications of exosomes involving ncRNAs, immune modulations, chemotherapeutic drugs, natural phytochemicals, small molecules, recombinant proteins, and the emerging concepts of fusogenic exosomes and vexosomes have been comprehensively reviewed which might be interesting realms of future research and therapeutic strategies”.

Round 2

Reviewer 2 Report

The authors have clarified several of the questions I raised in my previous review. Most of the major problems have been addressed by this revision